# SLIMMABLE NEURAL NETWORKS

**Jiahui Yu**[1]       **Linjie Yang**[2]       **Ning Xu**[2]       **Jianchao Yang**[3]       **Thomas Huang**[1]
[1]University of Illinois at Urbana-Champaign       [2]Snap Inc.       [3]ByteDance Inc.

## ABSTRACT

We present a simple and general method to train a single neural network executable at different widths[1], permitting instant and adaptive accuracy-efficiency trade-offs at runtime. Instead of training individual networks with different width configurations, we train a shared network with *switchable batch normalization*. At runtime, the network can adjust its width on the fly according to on-device benchmarks and resource constraints, rather than downloading and offloading different models. Our trained networks, named *slimmable neural networks*, achieve similar (and in many cases better) ImageNet classification accuracy than individually trained models of MobileNet v1, MobileNet v2, ShuffleNet and ResNet-50 at different widths respectively. We also demonstrate better performance of slimmable models compared with individual ones across a wide range of applications including COCO bounding-box object detection, instance segmentation and person keypoint detection without tuning hyper-parameters. Lastly we visualize and discuss the learned features of slimmable networks. Code and models are available at: https://github.com/JiahuiYu/slimmable_networks.

## 1    INTRODUCTION

Recently deep neural networks are prevailing in applications on mobile phones, augmented reality devices and autonomous cars. Many of these applications require a short response time. Towards this goal, manually designed lightweight networks (Howard et al., 2017; Zhang et al., 2017; Sandler et al., 2018) are proposed with low computational complexities and small memory footprints. Automated neural architecture search methods (Tan et al., 2018) also integrate on-device latency into search objectives by running models on a specific phone. However, at runtime these networks are not re-configurable to adapt across different devices given a same response time budget. For example, there were over 24,000 unique Android devices in 2015[2]. These devices have drastically different runtimes for the same neural network (Ignatov et al., 2018), as shown in Table 1. In practice, given the same response time constraint, high-end phones can achieve higher accuracy by running larger models, while low-end phones have to sacrifice accuracy to reduce latency.

Table 1: Runtime of MobileNet v1 for image classification on different devices.

|         | OnePlus 6 | Google Pixel | LG Nexus 5 | Samsung Galaxy S3 | ASUS ZenFone 2 |
|---------|-----------|--------------|------------|-------------------|----------------|
| Runtime | 24 ms     | 116 ms       | 332 ms     | 553 ms            | 1507 ms        |

Although a global hyper-parameter, width multiplier, is provided in lightweight networks (Howard et al., 2017; Zhang et al., 2017; Sandler et al., 2018) to trade off between latency and accuracy, it is inflexible and has many constraints. First, models with different width multipliers need to be trained, benchmarked and deployed individually. A big offline table needs to be maintained to document the allocation of different models to different devices, according to time and energy budget. Second, even on a same device, the computational budget varies (for example, excessive consumption of background apps reduces the available computing capacity), and the energy budget varies (for example, a mobile phone may be in low-power or power-saving mode). Third, when switching to a larger or smaller model, the cost of time and data for downloading and offloading models is not negligible.

---

[1]Width refers to number of channels in a layer.
[2]https://opensignal.com/reports/2015/08/android-fragmentation/

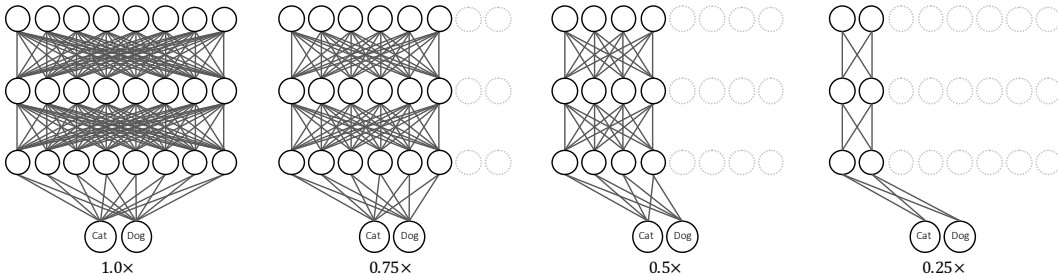

Figure 1: Illustration of slimmable neural networks. The same model can run at different widths (number of active channels), permitting instant and adaptive accuracy-efficiency trade-offs.

Recently dynamic neural networks are introduced to allow selective inference paths. Liu & Deng (2017) introduce controller modules whose outputs control whether to execute other modules. It has low theoretical computational complexity but is nontrivial to optimize and deploy on mobiles since dynamic conditions prohibit layer fusing and memory optimization. Huang et al. (2017) adapt early-exits into networks and connect them with dense connectivity. Wu et al. (2017) and Wang et al. (2017) propose to selectively choose the blocks in a deep residual network to execute during inference. Nevertheless, in contrast to width (number of channels), reducing depth cannot reduce memory footprint in inference, which is commonly constrained on mobiles.

The question remains: *Given budgets of resources, how to instantly, adaptively and efficiently trade off between accuracy and latency for neural networks at runtime?* In this work we introduce slimmable neural networks, a new class of networks executable at different widths, as a general solution to trade off between accuracy and latency on the fly. Figure 1 shows an example of a slimmable network that can switch between four model variants with different numbers of active channels. The parameters of all model variants are shared and the active channels in different layers can be adjusted. For brevity, we denote a model variant in a slimmable network as a *switch*, the number of active channels in a switch as its *width*. $0.25\times$ represents that the width in all layers are scaled by $0.25$ of the full model. In contrast to other solutions above, slimmable networks have several advantages: (1) For different conditions, a single model is trained, benchmarked and deployed. (2) A near-optimal trade-off can be achieved by running the model on a target device and adjusting active channels accordingly. (3) The solution is generally applicable to (normal, group, depthwise-separable, dilated) convolutions, fully-connected layers, pooling layers and many other building blocks of neural networks. It is also generally applicable to different tasks including classification, detection, identification, image restoration and more. (4) In practice, it is straightforward to deploy on mobiles with existing runtime libraries. After switching to a new configuration, the slimmable network becomes a normal network to run without additional runtime and memory cost.

However, neural networks naturally run as a whole and usually the number of channels cannot be adjusted dynamically. Empirically training neural networks with multiple switches has an extremely low testing accuracy around $0.1\%$ for 1000-class ImageNet classification. We conjecture it is mainly due to the problem that accumulating different number of channels results in different feature mean and variance. This discrepancy of feature mean and variance across different switches leads to inaccurate statistics of shared Batch Normalization layers (Ioffe & Szegedy, 2015), an important training stabilizer. To this end, we propose a simple and effective approach, *switchable batch normalization*, that privatizes batch normalization for different switches of a slimmable network. The variables of moving averaged means and variances can independently accumulate feature statistics of each switch. Moreover, Batch Normalization usually comes with two additional learnable scale and bias parameter to ensure same representation space (Ioffe & Szegedy, 2015). These two parameters may able to act as conditional parameters for different switches, since the computation graph of a slimmable network depends on the width configuration. It is noteworthy that the scale and bias can be merged into variables of moving mean and variance after training, thus by default we also use independent scale and bias as they come for free. Importantly, batch normalization layers usually have negligible size (less than $1\%$) in a model.

We first conduct comprehensive experiments on ImageNet classification task to show the effectiveness of switchable batch normalization for training slimmable neural networks. Compared with

individually trained networks, we demonstrate similar (and in many cases better) performances of slimmable MobileNet v1 $_{[0.25, 0.5, 0.75, 1.0]\times}$, MobileNet v2 $_{[0.35, 0.5, 0.75, 1.0]\times}$, ShuffleNet $_{[0.5, 1.0, 2.0]\times}$ and ResNet-50 $_{[0.25, 0.5, 0.75, 1.0]\times}$ ($_{[*]\times}$ denotes available switches). We further train a 8-switch slimmable MobileNet v1 $_{[0.25, 0.35, 0.45, 0.55, 0.65, 0.75, 0.85, 1.0]\times}$ without accuracy drop to demonstrate the scalability of our method. Beyond image classification, we also apply slimmable networks to various applications including COCO bounding-box object detection, instance segmentation and person keypoints detection. Experiments show that slimmable networks achieve better performance than individual ones at different widths respectively. The proposed slimmable networks are not only flexible and practical by design, but also effective, scalable and widely applicable according to our experiments. Lastly we visualize and discuss the learned features of slimmable networks.

## 2    RELATED WORK

**Model Pruning and Distilling.** Model pruning and distilling have a rich history in the literature of deep neural networks. Early methods (Han et al., 2015a;b) sparsify connections in neural networks. However, such networks usually require specific software and hardware accelerators to speedup. Driven by this fact, Molchanov et al. (2016), Wen et al. (2016), Li et al. (2016a), Liu et al. (2017), He et al. (2017), Luo et al. (2017), Anwar et al. (2017), Kim et al. (2017) and Ye et al. (2018) encourage structured sparsity by pruning channels, filters and network depth and fine-tuning iteratively with various penalty terms. As another family, model distilling methods (Hinton et al., 2015; Romero et al., 2014; Zhuang et al., 2018) first train a large network or an ensemble of networks, and then transfer the learned knowledge to a small model. Soft-targets and intermediate representations from trained large models are used to train a small model.

**Adaptive Computation Graph.** To reduce computation of a neural network, some works propose to adaptively construct the computation graph of a neural network. Liu & Deng (2017), Wu et al. (2017), Lin et al. (2017) and Wang et al. (2017) introduced additional controller modules or gating functions to determine the computation graph based on the current input. Amthor et al. (2016), Veit & Belongie (2017), Huang et al. (2017), Kuen et al. (2018) and Hu et al. (2017) implanted early-exiting prediction branches to reduce the average execution depth. The computation graph of these methods are conditioned on network input, and lower theoretical computational complexity can be achieved.

**Conditional Normalization.** Many real-world problems require conditional input. Feature-wise transformation (Dumoulin et al., 2018) is a prevalent approach to integrate different sources of information, where conditional scales and biases are applied across the network. It is commonly implemented in the form of conditional normalization layers, such as batch normalization or layer normalization (Ba et al., 2016). Conditional normalization is widely used in tasks including style transfer (Dumoulin et al., 2016; Li et al., 2017a; Huang & Belongie, 2017; Li et al., 2017b), image recognition (Li et al., 2016b; Yang et al., 2018) and many others (Perez et al., 2017b;a).

## 3    SLIMMABLE NEURAL NETWORKS

### 3.1    NAIVE TRAINING OR INCREMENTAL TRAINING

To train slimmable neural networks, we begin with a naive approach, where we directly train a shared neural network with different width configurations. The training framework is similar to the one of our final approach, as shown in Algorithm 1. The training is stable, however, the network obtains extremely low top-1 testing accuracy around $0.1\%$ on 1000-class ImageNet classification. Error curves of the naive approach are shown in Figure 2. We conjecture the major problem in the naive approach is that: for a single channel in a layer, different numbers of input channels in previous layer result in different means and variances of the aggregated feature, which are then rolling averaged to a shared batch normalization layer. The inconsistency leads to inaccurate batch normalization statistics in a layer-by-layer propagating manner. Note that these batch normalization statistics (moving averaged means and variances) are only used during testing, in training the means and variances of the current mini-batch are used.

We then investigate incremental training approach (a.k.a. progressive training) (Tann et al., 2016). We experiment with Mobilenet v2 on ImageNet classification task. We first train a base model $A$

(MobileNet v2 $0.35\times$). We fix it and add extra parameters $B$ to make it an extended model $A+B$ (MobileNet v2 $0.5\times$). The extra parameters are fine-tuned along with the fixed parameters of $A$ on the training data. Although the approach is stable in both training and testing, the top-1 accuracy only increases from $60.3\%$ of $A$ to $61.0\%$ of $A+B$. In contrast, individually trained MobileNet v2 $0.5\times$ achieves $65.4\%$ accuracy on the ImageNet validation set. The major reason for this accuracy degradation is that when expanding base model $A$ to the next level $A+B$, new connections, not only from $B$ to $B$, but also from $B$ to $A$ and from $A$ to $B$, are added in the computation graph. The incremental training prohibits joint adaptation of weights $A$ and $B$, significantly deteriorating the overall performance.

## 3.2 SWITCHABLE BATCH NORMALIZATION

Motivated by the investigations above, we present a simple and highly effective approach, named *Switchable Batch Normalization* (*S-BN*), that employs independent batch normalization (Ioffe & Szegedy, 2015) for different switches in a slimmable network. Batch normalization (BN) was originally proposed to reduce internal covariate shift by normalizing the feature: $y' = \gamma \frac{y-\mu}{\sqrt{\sigma^2+\epsilon}} + \beta$, where $y$ is the input to be normalized and $y'$ is the output, $\gamma, \beta$ are learnable scale and bias, $\mu, \sigma^2$ are mean and variance of current mini-batch during training. During testing, moving averaged statistics of means and variances across all training images are used instead. BN enables faster and stabler training of deep neural networks (Ioffe & Szegedy, 2015; Radford et al., 2015), also it can encode conditional information to feature representations (Perez et al., 2017b; Li et al., 2016b).

To train slimmable networks, *S-BN* privatizes all batch normalization layers for each switch in a slimmable network. Compared with the naive training approach, it solves the problem of feature aggregation inconsistency between different switches by independently normalizing the feature mean and variance during testing. The scale and bias in *S-BN* may be able to encode conditional information of width configuration of current switch (the scale and bias can be merged into variables of moving mean and variance after training, thus by default we also use independent scale and bias as they come for free). Moreover, in contrast to incremental training, with *S-BN* we can jointly train all switches at different widths, therefore all weights are jointly updated to achieve a better performance. A representative training and validation error curve with *S-BN* is shown in Figure 2.

*S-BN* also has two important advantages. First, the number of extra parameters is negligible. Table 2 enumerates the number and percentage of parameters in batch normalization layers (after training, $\mu, \sigma, \gamma, \beta$ are merged into two parameters). In most cases, batch normalization layers only have less than $1\%$ of the model size. Second, the runtime overhead is also negligible for deployment. In practice, batch normalization layers are typically fused into convolution layers for efficient inference. For slimmable networks, the re-fusing of batch normalization can be done on the fly at runtime since its time cost is negligible. After switching to a new configuration, the slimmable network becomes a normal network to run without additional runtime and memory cost.

Table 2: Number and percentage of parameters in batch normalization layers.

|  | MobileNet v1 $1.0\times$ | MobileNet v2 $1.0\times$ | ShuffleNet $2.0\times$ | ResNet-50 $1.0\times$ |
|---|---|---|---|---|
| Conv and FC | 4,210,088 (99.483%) | 3,470,760 (99.027%) | 5,401,816 (99.102%) | 25,503,912 (99.792%) |
| BatchNorm | 21,888 (0.517%) | 34,112 (0.973%) | 48,960 (0.898%) | 53,120 (0.208%) |

## 3.3 TRAINING SLIMMABLE NEURAL NETWORKS

Our primary objective to train a slimmable neural network is to optimize its accuracy averaged from all switches. Thus, we compute the loss of the model by taking an un-weighted sum of all training losses of different switches. Algorithm 1 illustrates a memory-efficient implementation of the training framework, which is straightforward to integrate into current neural network libraries. The *switchable width list* is predefined, indicating the available switches in a slimmable network. During training, we accumulate back-propagated gradients of all switches, and update weights afterwards. Empirically we find no hyper-parameter needs to be tuned specifically in all of our experiments.

---

**Algorithm 1** Training slimmable neural network $M$.

---

**Require:** Define *switchable width list* for slimmable network $M$, for example, $[0.25, 0.5, 0.75, 1.0] \times$.
 1: Initialize shared convolutions and fully-connected layers for slimmable network $M$.
 2: Initialize independent batch normalization parameters for each *width* in *switchable width list*.
 3: **for** $i = 1, ..., n_{iters}$ **do**
 4:     Get next mini-batch of data $x$ and label $y$.
 5:     Clear gradients of weights, $optimizer.zero\_grad()$.
 6:     **for** *width* in *switchable width list* **do**
 7:         Switch the batch normalization parameters of current width on network $M$.
 8:         Execute sub-network at current width, $\hat{y} = M'(x)$.
 9:         Compute loss, $loss = criterion(\hat{y}, y)$.
10:         Compute gradients, $loss.backward()$.
11:     **end for**
12:     Update weights, $optimizer.step()$.
13: **end for**

---

## 4   EXPERIMENTS

In this section, we first evaluate slimmable networks on ImageNet (Deng et al., 2009) classification. Further we demonstrate the performance of a slimmable network with more switches. Finally we apply slimmable networks to a number of different applications.

### 4.1   IMAGENET CLASSIFICATION

We experiment with the ImageNet (Deng et al., 2009) classification dataset with 1000 classes. It is comprised of around 1.28M training images and 50K validation images.

We first investigate slimmable neural networks on three state-of-the-art lightweight networks, MobileNet v1 (Howard et al., 2017), MobileNet v2 (Sandler et al., 2018), ShuffleNet (Zhang et al., 2017), and one representative large model ResNet-50 (He et al., 2016).

To make a fair comparison, we follow the training settings (for example, learning rate scheduling, weight initialization, weight decay, data augmentation, input image resolution, mini-batch size, training iterations, optimizer) in corresponding papers respectively (Howard et al., 2017; Sandler et al., 2018; Zhang et al., 2017; He et al., 2016). One exception is that for MobileNet v1 and MobileNet v2, we use stochastic gradient descent (SGD) as the optimizer instead of the RMSPropOptimizer (Howard et al., 2017; Sandler et al., 2018). For ResNet-50 (He et al., 2016), we train for 100 epochs, and decrease the learning rate by $10\times$ at 30, 60 and 90 epochs. We evaluate the top-1 classication error on the center $224 \times 224$ crop of images in the validation set. More implementation details are included in Appendix A.

We first show training and validation error curves in Figure 2. The results of naive training approach are also reported as comparisons. Although both our approach and the naive approach are stable in training, the testing error of naive approach is extremely high. With switchable batch normalization, the error rates of different switches are stable and the rank of error rates is also preserved consistently across all training epochs.

Next we show in Table 3 the top-1 classification error for both individual networks and slimmable networks given same width configurations. We use S- to indicate slimmable models. The error rates for individual models are from corresponding papers except those denoted with [†]. The runtime FLOPs (number of Multiply-Adds) for each model are also reported as a reference. Table 3 shows that slimmable networks achieve similar performance compared to those that are individually trained. Intuitively compressing different networks into a shared network poses extra optimization constraints to each network, a slimmable network is expected to have lower performance than individually trained ones. However, our experiments show that joint training of different switches indeed improves the performance in many cases, especially for slim switches (for example, MobileNet v1 $0.25\times$ is improved by 3.3%). We conjecture that the improvements may come from implicit model distilling (Hinton et al., 2015; Romero et al., 2014) where the large model transfers its knowledge to small model by weight sharing and joint training.

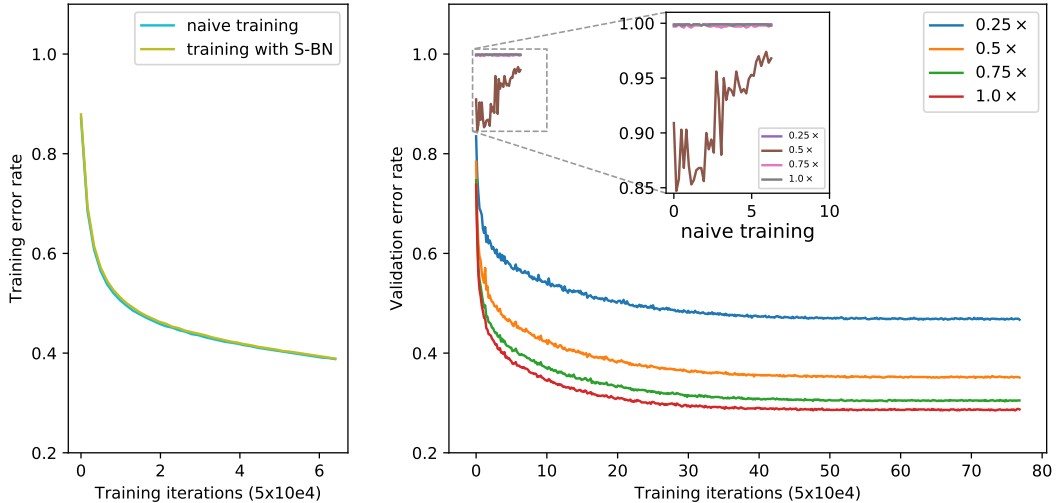

Figure 2: Training and validation curves of slimmable networks. Left shows the training error of the largest switch. Right shows testing errors on validation set with different switches. For naive approach, the training is stable (left) but testing error is high (right, zoomed). Slimmable networks trained with *S-BN* have stable and rank-preserved testing accuracy across all training iterations.

Table 3: Results of ImageNet classification. We show top-1 error rates of individually trained networks and slimmable networks given same width configurations and FLOPs. We use S- to indicate slimmable models, [†] to denote our reproduced result.

| Individual Networks | | | Slimmable Networks | | | FLOPs |
|---|---|---|---|---|---|---|
| Name | Params | Top-1 Err. | Name | Params | Top-1 Err. | |
| MobileNet v1 $1.0\times$ | 4.2M | 29.1 | S-MobileNet v1 $[0.25, 0.5, 0.75, 1.0]\times$ | 4.3M | 28.5 (0.6) | 569M |
| MobileNet v1 $0.75\times$ | 2.6M | 31.6 | | | 30.5 (1.1) | 317M |
| MobileNet v1 $0.5\times$ | 1.3M | 36.7 | | | 35.2 (1.5) | 150M |
| MobileNet v1 $0.25\times$ | 0.5M | 50.2 | | | 46.9 (3.3) | 41M |
| MobileNet v2 $1.0\times$ | 3.5M | 28.2 | S-MobileNet v2 $[0.35, 0.5, 0.75, 1.0]\times$ | 3.6M | 29.5 (-1.3) | 301M |
| MobileNet v2 $0.75\times$ | 2.6M | 30.2 | | | 31.1 (-0.9) | 209M |
| MobileNet v2 $0.5\times$ | 2.0M | 34.6 | | | 35.6 (-1.0) | 97M |
| MobileNet v2 $0.35\times$ | 1.7M | 39.7 | | | 40.3 (-0.6) | 59M |
| ShuffleNet $2.0\times$ | 5.4M | 26.3 | S-ShuffleNet $[0.5, 1.0, 2.0]\times$ | 5.5M | 28.7 (-2.4) | 524M |
| ShuffleNet $1.0\times$ | 1.8M | 32.6 | | | 34.5 (-0.9) | 138M |
| ShuffleNet $0.5\times$ | 0.7M | 43.2 | | | 42.7 (0.5) | 38M |
| ResNet-50 $1.0\times$ | 25.5M | 23.9 | S-ResNet-50 $[0.25, 0.5, 0.75, 1.0]\times$ | 25.6M | 24.0 (-0.1) | 4.1G |
| ResNet-50 $0.75\times$[†] | 14.7M | 25.3 | | | 25.1 (0.2) | 2.3G |
| ResNet-50 $0.5\times$[†] | 6.9M | 28.0 | | | 27.9 (0.1) | 1.1G |
| ResNet-50 $0.25\times$[†] | 2.0M | 36.2 | | | 35.0 (1.2) | 278M |

Our proposed approach for slimmable neural networks is generally applicable to the above representative network architectures. It is noteworthy that we experiment with both residual and non-residual networks (MobileNet v1). The training of slimmable models can be applied to convolutions, depthwise-separable convolutions (Chollet, 2016), group convolutions (Xie et al., 2017), pooling layers, fully-connectted layers, residual connections, feature concatenations and many other building blocks of deep neural networks.

## 4.2 More Switches in Slimmable Networks

The more switches available in a slimmable network, the more choices one have for trade-offs between accuracy and latency. We thus investigate how the number of switches potentially impact accuracy. In Table 4, we train a 8-switch slimmable MobileNet v1 and compare it with 4-switch and individually trained ones. The results show that a slimmable network with more switches have similar performance, demonstrating the scalability of our proposed approach.

Table 4: Top-1 error rates on ImageNet classification with individually trained networks, 4-switch S-MobileNet v1 $_{[0.25, 0.5, 0.75, 1.0]\times}$ and 8-switch S-MobileNet v1 $_{[0.25, 0.35, 0.45, 0.55, 0.65, 0.75, 0.85, 1.0]\times}$.

| | $0.25\times$ | $0.35\times$ | $0.45\times$ | $0.5\times$ | $0.55\times$ | $0.65\times$ | $0.75\times$ | $0.85\times$ | $1.0\times$ |
|---|---|---|---|---|---|---|---|---|---|
| Individual | 50.2 | - | - | 36.7 | - | - | 31.6 | - | 29.1 |
| 4-switch | 46.9 $_{(3.3)}$ | - | - | 35.2 $_{(1.5)}$ | - | - | 30.5 $_{(1.1)}$ | - | 28.5 $_{(0.6)}$ |
| 8-switch | 47.6 $_{(2.6)}$ | 41.1 | 36.6 | - | 33.8 | 31.4 | 30.2 $_{(1.4)}$ | 29.2 | 28.4 $_{(0.7)}$ |

## 4.3 Object Detection, Instance Segmentation and Keypoints Detection

Finally, we apply slimmable networks on tasks of bounding-box object detection, instance segmentation and keypoints detection based on detection frameworks MMDetection (Chen et al., 2018) and Detectron (Girshick et al., 2018).

Table 5: Average precision (AP) on COCO 2017 validation set with individually trained networks and slimmable networks. ResNet-50 models are used as backbones for Faster-RCNN, Mask-RCNN and Keypoints-RCNN based on detection frameworks (Girshick et al., 2018; Chen et al., 2018). Faster $1.0\times$ indicates Faster-RCNN for object detection with ResNet-50 $1.0\times$ as backbone.

| Type | Individual Networks | | | Slimmable Networks | | |
|---|---|---|---|---|---|---|
| | Box AP | Mask AP | Kps AP | Box AP | Mask AP | Kps AP |
| Faster $1.0\times$ | 36.4 | - | - | 36.8 $_{(0.4)}$ | - | - |
| Faster $0.75\times$ | 34.7 | - | - | 36.1 $_{(1.4)}$ | - | - |
| Faster $0.5\times$ | 32.7 | - | - | 34.0 $_{(1.3)}$ | - | - |
| Faster $0.25\times$ | 27.5 | - | - | 29.6 $_{(2.1)}$ | - | - |
| Mask $1.0\times$ | 37.3 | 34.2 | - | 37.4 $_{(0.1)}$ | 34.9 $_{(0.7)}$ | - |
| Mask $0.75\times$ | 35.6 | 32.9 | - | 36.7 $_{(1.1)}$ | 34.3 $_{(1.4)}$ | - |
| Mask $0.5\times$ | 33.4 | 30.9 | - | 34.7 $_{(1.5)}$ | 32.6 $_{(1.7)}$ | - |
| Mask $0.25\times$ | 28.2 | 26.6 | - | 30.2 $_{(2.0)}$ | 28.6 $_{(2.0)}$ | - |
| Kps $1.0\times$ | 50.5 | - | 61.3 | 52.8 $_{(2.3)}$ | - | 63.9 $_{(2.6)}$ |
| Kps $0.75\times$ | 49.6 | - | 60.5 | 52.7 $_{(3.1)}$ | - | 63.6 $_{(3.1)}$ |
| Kps $0.5\times$ | 48.5 | - | 59.8 | 51.6 $_{(3.1)}$ | - | 62.6 $_{(2.8)}$ |
| Kps $0.25\times$ | 45.4 | - | 56.7 | 48.2 $_{(2.8)}$ | - | 59.5 $_{(2.8)}$ |

Following the settings of *R-50-FPN*-$1\times$ (Lin et al., 2016; Girshick et al., 2018; Chen et al., 2018), pre-trained ResNet-50 models at different widths are fine-tuned and evaluated. The lateral convolution layers in feature pyramid network (Lin et al., 2016) are same for different pre-trained backbone networks. For individual models, we train ResNet-50 with different width multipliers on ImageNet and fine-tune them on each task individually. For slimmable models, we first train on ImageNet using Algorithm 1. Following Girshick et al. (2018), the moving averaged means and variances of switchable batch normalization are also fixed after training. Then we fine-tune the slimmable models on each task using Algorithm 1. The detection head and lateral convolution layers in feature pyramid network (Lin et al., 2016) are shared across different switches in a slimmable network. In this way, each switch in a slimmable network is with exactly same network architecture and FLOPs with its individual baseline. More details of implementation are included in Appendix B. We train all models on COCO 2017 train set and report Average Precision (AP) on COCO 2017 validation set in Table 5. In general, slimmable neural networks perform better than individually trained ones, especially for slim network architectures. The gain of performance is presumably due to implicit model distillation (Hinton et al., 2015; Romero et al., 2014) and richer supervision signals.

# 5 VISUALIZATION AND DISCUSSION

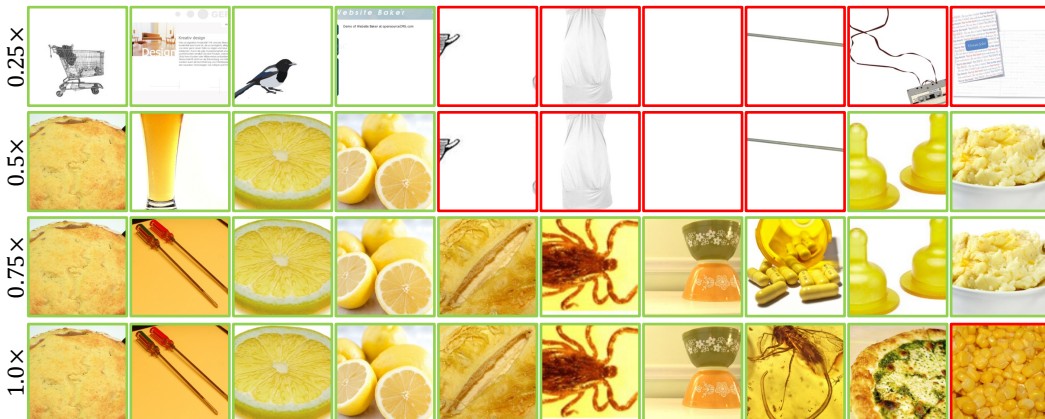

Figure 3: Top-activated images for same channel 3_9 in different switches in S-MobileNet v1. Different rows represent results from different switches. Images with red outlines are mis-classified. Note that the white color in RGB is $[255, 255, 255]$, yellow in RGB is $[255, 255, 0]$.

**Visualization of Top-activated Images.** Our primary interest lies in understanding the role that the same channel played in different switches in a slimmable network. We employ a simple visualization approach (Girshick et al., 2014) to visualize the images with highest activation values on a specific channel. Figure 3 shows the top-activated images of the same channel in different switches. Images with green outlines are correctly classified by the corresponding model, while images with red outlines are mis-classified. Interestingly the results show that for different switches, the major role of same channel (channel 3_9 in S-MobileNet v1) transits from recognizing white color (RGB value $[255, 255, 255]$) to yellow color (RGB value $[255, 255, 0]$) when the network width increases. It indicates that the same channel in slimmable network may play similar roles (in this case to recognize colors of RGB value $[255, 255, *]$) but have slight variations in different switches (the one in quarter-sized model focuses more on white color while the one in full model on yellow color).

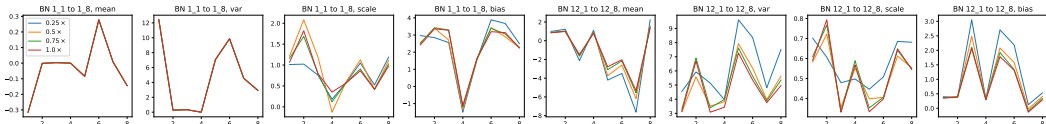

Figure 4: Values of BN parameters in different switches. We show BN values of both shallow (left, BN 1_1 to 1_8) and deep (right, BN 12_1 to 12_8) layers of S-MobileNet v1.

**Values of Switchable Batch Normalization.** Our proposed *S-BN* learns different BN transformations for different switches. But how diverse are the learned BN parameters? We show the values of batch normalization weights in both shallow (BN 1_1 to 1_8) and deep (BN 12_1 to 12_8) layers of S-MobileNet v1 in Figure 4. The results show that for shallow layers, the mean, variance, scale and bias are very close, while in deep layers they are diverse. The value discrepancy is increased layer by layer in our observation, which also indicates that the learned features of a same channel in different switches have slight variations of semantics.

# 6 CONCLUSION

We introduced slimmable networks that permit instant and adaptive accuracy-efficiency trade-offs at runtime. Switchable batch normalization is proposed to facilitate robust training of slimmable networks. Compared with individually trained models with same width configurations, slimmable networks have similar or better performances on tasks of classification, object detection, instance segmentation and keypoints detection. The proposed slimmable networks and slimmable training could be further applied to unsupervised learning and reinforcement learning, and may help to related fields such as network pruning and model distillation.

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

## A  TRAINING ON IMAGENET

We mainly use three training settings corresponding to Howard et al. (2017); Sandler et al. (2018); Zhang et al. (2017); He et al. (2016). For MobileNet v1 and MobileNet v2, we train 480 epochs with mini-batch size 160, and exponentially ($\gamma = 0.98$) decrease learning rate starting from 0.045 per epoch. For ShuffleNet ($g = 3$), we train 250 epochs with mini-batch size 512, and linearly decrease learning rate from 0.25 to 0 per iteration. For ResNet-50, we train 100 epochs with mini-batch size 256, and decrease the learning rate by $10\times$ at 30, 60 and 90 epochs. We use stochastic gradient descent (SGD) as optimizer, Nesterov momentum with a momentum weight of 0.9 without dampening, and a weight decay of $10^{-4}$ for all training settings. All models are trained on 4 Tesla P100 GPUs and the batch mean and variance of batch normalization are computed within each GPU.

With the above training settings, the reproduced MobileNet v1 $1.0\times$, MobileNet v2 $1.0\times$ and ResNet-50 $1.0\times$ have similar top-1 accuracy ($\pm0.5\%$). Our reproduced ShuffleNet $2.0\times$ has top-1 error rate 28.2%, which is 1.9% worse than results in Zhang et al. (2017). It is likely due to the inconsistency of mini-batch size and number of training GPUs.

## B  TRAINING ON COCO

We use pytorch-style ResNet-50 model (Chen et al., 2018) as backbone for COCO tasks, since our pretrained ResNet-50 at different widths for ImageNet classification is also pytorch-style. However, it is slightly different than the caffe-style ResNet-50 used in Detectron (Girshick et al., 2018) (the stride for down-sampling is added into $3 \times 3$ convolutions instead of $1 \times 1$ convolutions). To this

end, we mainly conduct COCO experiments based on another detection framework: MMDetection (Chen et al., 2018), which has hyper-parameter settings with same pytorch-style ResNet-50. With same hyper-parameter settings (i.e., $RCNN\_R50\_FPN\_1\times$), we fine-tune both individual ResNet-50 models and slimmable ResNet-50 on tasks of object detection and instance segmentation. Our reproduced results on ResNet-50 $1.0\times$ is consistent with official models in MMDetection (Chen et al., 2018). For keypoint detection task, we conduct experiment on Detectron (Girshick et al., 2018) framework by modifying caffe-style ResNet-50 to pytorch-style and training on 4 GPUs without other modification of hyper-parameters. We have released code (training and testing) and pretrained models on both ImageNet classification task and COCO detection tasks.

## C    ABLATION STUDY OF CONDITIONAL PARAMETERS IN BN

In our work, private parameters $\gamma$, $\beta$, $\mu$, $\sigma^2$ of BN are introduced in *Switchable Batch Normalization* for each sub-network to independently normalize feature $y' = \gamma\frac{y-\mu}{\sqrt{\sigma^2+\epsilon}} + \beta$, where $y$ is input and $y'$ is output, $\gamma$, $\beta$ are learnable scale and bias, $\mu$, $\sigma^2$ are moving averaged statistics for testing. In switchable batch normalization, the private $\gamma$, $\beta$ come for free because after training, they can be merged as $y' = \gamma' y + \beta', \gamma' = \frac{\gamma}{\sqrt{\sigma^2+\epsilon}}, \beta' = \beta - \gamma'\mu$. Nevertheless, we present ablation study on how these conditional parameters affect overall performance. The results are shown in Table 6.

Table 6: Top-1 error rates on ImageNet classification with two S-MobileNet v1 $[0.25, 0.5, 0.75, 1.0]\times$ with private scale and bias or shared ones.

|                        | $0.25\times$ | $0.5\times$ | $0.75\times$ | $1.0\times$ |
|------------------------|--------------|-------------|--------------|-------------|
| Private $\gamma$, $\beta$ | 46.9         | 35.2        | 30.5         | 28.5        |
| Shared $\gamma$, $\beta$  | 47.1 (-0.2)  | 35.9 (-0.7) | 30.9 (-0.4)  | 28.8 (-0.3) |

