# OpenReview forum: "Slimmable Neural Networks"
_ICLR.cc/2019/Conference_

### Official Review · AnonReviewer3 · 2018-11-03
**algo details and numbers**

**Rating:** 7
**Confidence:** 4

**Review:**

This paper trains a single network executable at different widths. This is implemented by maintaining separate BN parameter and statistics for different width. The problem is well-motivated and the proposed method can be very helpful for deployment of deep models to devices with varying capacity and computational ability.

This paper is well-written and the experiments are performed on various structures. Still I have several concerns regarding the algorithm.
1. In algo 1, while gradients for convolutional and fully-connected layers are accumulated for all switches before update, how are the parameters for different switches updated?
2. In algo 1, the gradients of all switches are accumulated before the update. This may result in implicit unbalanced gradient information, e.g. the connections in 0.25x model in Figure 1 has gradient flows on all four different switches,  while the right-most 0.25x connections in 1.0x model has only one gradient flow from the 1.0x switch, will this unbalanced gradient information increase optimization difficulty and how is it solved?
3.  In the original ResNet paper, https://arxiv.org/pdf/1512.03385.pdf, the top-1 error of RestNet-50 is <21% in Table 4. The number reported in this paper (Table 3) is 23.9. Where does the difference come from?

---

> ### Author Response · Authors · 2018-11-18
> **Authors' Reply to Review**
>
> Thanks for your review efforts! We have addressed all three questions below:
>
> 1. As mentioned in Section 3.3, the only modification is to accumulate all gradients from different switches. It means that the optimizer (SGD for image recognition tasks) is exactly the same as training individual models (same momentum, etc.). The only difference is the value of gradient for each parameter. In Algorithm 1, we follow pytorch-style API and use optimizer.step() to indicate applying gradients. We have not observed any difficulty in optimization of slimmable networks using default optimizer in Algorithm 1.
>
> 2. There is no "unbalanced gradient" problem in training slimmable networks (it may seem like so). The parameters of 0.25x seem to have "more gradients", but in the forward view, these parameters of 0.25x are also used four times in Net 0.25x, 0.5x, 0.75x and 1.0x. It means the parameters in 0.25x are more important for the overall performance of slimmable networks. In fact, back-propagation is strictly based on forward feature propagation. In the forward view, as mentioned in Section 3.3, our primary objective to train a slimmable network is to optimize its accuracy averaged from all switches.
>
> 3. Our reported ResNet-50 accuracy is correct (23.9 top-1 error). We evaluate single-crop testing accuracy instead of 10-crop following all our baselines. The ResNet-50 single-crop testing accuracy is publicly reported in ResNeXt paper (Table 3, 1st row) [1], released code [2] and many other publications. Our ResNet-50 has same implementation with PyTorch official pre-trained model zoo [3] where the top-1 error is also 23.9 instead of <21% (in fact ResNet-152 still has > 21% single-crop top-1 error rate).
>
> We sincerely hope the rating can be reconsidered if it was affected by above questions. Thanks for your time and review efforts!
>
>
> [1] Xie, Saining, et al. "Aggregated residual transformations for deep neural networks." Computer Vision and Pattern Recognition (CVPR), 2017 IEEE Conference on. IEEE, 2017.
> [2] https://github.com/facebookresearch/ResNeXt
> [3] https://pytorch.org/docs/stable/torchvision/models.html

---

> > ### Comment · AnonReviewer3 · 2018-11-27
> > **Thanks for addressing my questions**
> >
> > Thanks for addressing my questions!

---

### Official Review · AnonReviewer1 · 2018-11-05
**Very exciting work**

**Rating:** 9
**Confidence:** 5

**Review:**

This paper presents a straightforward looking approach for creating a neural networks that can run under different resource constraints, e.g. less computation but lower quality solution and expensive high quality solution, while all the networks are having the same filters. The idea is to share the filters of the cheapest network with those of the larger more expensive networksa and train all those networks jointly with weight sharing. One important practical observation is that the batch-normalization parameters should not be shared between those filters in order to get good results. However, the most interesting surprising observation, that is the main novelty of the work that even the highest quality vision network get substantially better by this training methodology as compared to be training alone without any weight sharing with the smaller networks, when trained for object detection and segmentation purposes (but not for recognition). This is a highly unexpected result and provides a new unanticipated way of training better segmentation models. It is especially nice that the paper does not pretend that this phenomenon is well understood but leaves its proper explanation for future work. I think a lot of interesting work is to be expected along these lines.

---

> ### Author Response · Authors · 2018-11-18
> **Authors' Reply to Review**
>
> Thanks for your positive review and encouragements! We also believe the discovery of slimmable network opens up the possibility to many related fields including model distillation, network compression and better representation learning. We are actively exploring on these topics and hope this submission may contribute to ICLR community.

---

### Official Review · AnonReviewer2 · 2018-11-05
**The paper proposes an idea of combining different size models together into one shared net. And the performance is claimed to be slightly worse for classification and much better for detection.**

**Rating:** 8
**Confidence:** 4

**Review:**

The idea is really interesting. One only need to train and maintain one single model, but use it in different platforms of different computational power.

And according to the experiment results of COCO detection, the S-version models are much better than original versions (eg. faster-0.25x, from 24.6 to 30.0) . The improvement is huge to me. However the authors do not explain any deep reasons.

And for classification, there are slightly performance drop instead of a large improvement which is also hard to understand.

For detection, experiments on depth-wise convolution based models (such as mobilenet and shufflenet) are suggested to make this work more solid and meaningful.

---

> ### Author Response · Authors · 2018-11-18
> **Authors' Reply to Review**
>
> Thanks for your review efforts! We have addressed all questions below:
>
> 1. We aim to train single neural network executable at different widths. We find slimmable networks achieve better results especially for small models (e.g., 0.25x) on detection tasks. We have mentioned that it is probably due to implicit distillation, richer supervision and better learned representation (since detection results are based on pre-trained ImageNet learned representation). We try to avoid strong claims of any deep reason because none of them is strictly proved by us yet. Explaining deep reasons for improvements are not the motivation or the focus of this paper. But we are actively exploring on these questions!
>
> 2. In fact, on average the image classification results are also improved (0.5 better top-1 accuracy in total), especially for small models. After submission, we have improved accuracy of S-ShuffleNet due to an additional ReLU layer (our implementation bug) between depthwise convolution and group convolution (Figure 2 of ShuffleNet [3]). Our models will be released.
>
> 3. Thanks for the good suggestion! Currently we conduct detection experiments mainly on Detectron [1] and MMDetection [2] framework where ResNet-50 is among the most efficient models. We do value this suggestion and will try to implement mobilenet-based detectors. Besides, all code (including classification and detection) and pre-trained models will be released soon and we warmly welcome the community to work on together.
>
> Thanks!
>
>
> [1] https://github.com/facebookresearch/Detectron
> [2] https://github.com/open-mmlab/mmdetection
> [3] Zhang et al. Shufflenet: An extremely efficientconvolutional neural network for mobile devices.arXiv preprint arXiv:1707.01083, 2017.

---

### Public Comment · ~Jason_Kuen1 · 2018-10-29
**A related work**

Nice work! I had a paper published at CVPR 2018 on training convolutional networks that support instant and adaptive accuracy-efficiency trade-offs at runtime, via early downsampling rather than networking slimming. My paper also includes a similar technique of using independent BatchNorm parameters (just means and stds in my paper, whereas you "unshare" all of BatchNorm parameters) for different trade-off configurations.

I'd appreciate if you would include a reference to it - "Stochastic Downsampling for Cost-Adjustable Inference and Improved Regularization in Convolutional Networks". Thanks.

---

> ### Author Response · Authors · 2018-11-18
> **Authors' Reply to Comment**
>
> Thanks for your interest in our work! We have added the citation.

---

### Public Comment · (anonymous) · 2018-11-07
**Good motivation but not convincing results**

The motivation to train one model end deploy in multiple devices is quite interesting.  However, the experimental results are not convincing.

In Table 3, most of the S-networks reduce performance compared to their individual counterparts. It's not cumbersome to train individual slimmed model that has higher accuracy in portable device and the same FLOPs as the S-model, since training runtime is not the key problem with increasing amount of computational powers.

In Table 5, the baselines of R-50-FPN-1× are much lower than those reported in the original paper of Faster R-CNN and Mask R-CNN.  In previous work, the box and mask AP of Mask+R-50-FPN-1× are 37.3 and 33.7, while box AP for Faster+R-50-FPN-1× is 36.4. These results are already comparable and even better than the S-networks. The same problem applies to the keypoints. Therefore, it is unclear that S-model would still bring performance gain when the standard baselines are employed.

Another concern is that S-model seems to degenerate performance in ImageNet, as the paper mentioned "a slimmable network is expected to have lower performance than individually trained ones intuitively". But it turns out that the pretrained S-model in ImageNet has large improvement when finetuned in detection and segmentation. This violates common sense.

---

> ### Author Response · Authors · 2018-11-18
> **Authors' Reply to Comment**
>
> Thanks for your interest in our work! However, we cannot agree with your comments. We have addressed your questions and concerns below:
>
> 1. As introduced in Sec. 1 and concluded by all reviewers, this work aims to "train a single neural network executable at different widths for different devices"
> We never claim "training runtime is the key problem". And our focus is not on "training a single network" but on "a single network executable at different widths". The testing runtime and flexible accuracy-efficiency trade-offs are what we care.
> 2. In Table 3 for ImageNet classification, the top-1 accuracy is actually improved by 0.5 in total.
>
> 3. Although all experiments are conducted with same settings for both individual and slimmable models, we also noticed that the reproduced performance of individual models was lower than original papers. A potential reason is included in Appendix B of the first submitted version (original *-RCNN papers use ResNet-50 with strides on 1x1 convolution, while we follow PyTorch official implemented ResNet-50 with strides on 3x3). After submission, we found a recently released detection framework MMDetection [1] that has settings for pytorch-style ResNet-50. Thus we have conducted another set of detection experiments and included the results in Appendix C (same mAP is reproduced, for example, Faster-R-50-FPN-1x with 36.4 mAP).
> And our conclusion still holds: on detection tasks, slimmable models have better performance than individually trained models, especially for small models. Specifically, for 0.25x models, slimmable network has 2.0+ mAP, which is indeed significant. For 1.0x models, slimmable also have 0.4+ mAP, 0.7+ mAP for Faster-RCNN and Mask-RCNN. We will fully release our code (both training and testing) and pre-trained models on both ImageNet classification and COCO detection sets.
>
> 4. Image classification trains models from scratch, while COCO detection fine-tunes pre-trained ImageNet models. The improvement on detection may due to better learned representation of slimmable models on ImageNet when transfer to COCO tasks. We have also mentioned in our submission that it is probably due to implicit distillation and richer supervision. The reason behind the improvements is beyond the motivation of this submission and requires future investigation. We try to avoid strong claims of any deep reason because none of them is strictly proved by us yet.
>
> We sincerely thank you for posting these concerns and we will always try our best to address them. Please let us know if you have further question or concern. Thanks!
>
>
> [1] https://github.com/open-mmlab/mmdetection

---

### Public Comment · (anonymous) · 2018-11-12
**"reducing depth cannot reduce memory footprint"?**

It is claimed in the 3rd paragraph in introduction that,

 "Nevertheless, in contrast to width (number of channels), reducing depth cannot reduce memory footprint which is commonly constrained during runtime."

However, in my understanding, the momory reduces linearly when reducing depth for deep neural network. Could you please explain more on this?

---

> ### Author Response · Authors · 2018-11-18
> **Authors' Reply to Comment**
>
> Thanks for your interest in our work. Our claim is correct: at runtime reducing depth cannot reduce memory footprint.
>
> For a simple example, consider a layer-by-layer network stacking same convolution layers, the output of layer N can always be placed into the memory of its input after computation, and feed into next layer (N+1). Because at runtime, there is no need to store feature of previous layers generally (in training, they are required for gradient computation).
>
> A good reference is MobileNet v2 paper [1], section 5.1 memory efficient inference. It shows that the memory footprint can be simplified to: M = max_{layer_i \in all layers} (memory_input of layer_i + memory_output of layer_i).
>
> The memory footprint M is a MAX operation over all layers, instead of SUM, during inference.
>
>
> [1] Sandler, Mark, et al. “MobileNet v2: Inverted residuals and linear bottlenecks: Mobile networks for classification, detection and segmentation." arXiv preprint arXiv:1801.04381 (2018).

---

### Public Comment · ~Ji_Lin1 · 2018-11-27
**Interesting work and a related paper**

Very interesting work and congratulations!

I am the first author of paper Runtime Neural Pruning (RNP, in NIPS 2017), where we also partitioned the channels of each convolutional layers into 4 equal sets and used a reinforcement learning agent to determine how many sets to run according to the difficulty of input images, in an incremental way. RNP can also adjust the workload according to the available hardware resources by adjusting the computation penalty.

I think your paper has solved some of the training difficulty in RNP, and it would be very interesting to try a network-level dynamic inference according to the input image. Also, it would be very nice if you can include a reference to our paper. Thanks!

---

> ### Author Response · Authors · 2018-11-27
> **Authors' Reply to Comment**
>
> Thanks for your interest in our work! We will add the citation once revision period is re-opened. Code will be released soon and we warmly welcome the community to work together on related topics!

---

### Public Comment · (anonymous) · 2018-11-30
**Missing related work**

This paper introduces a deep neural network that provides different inference paths with respect to different widths for accuracy-efficiency trade-off at test time, but the concept has been already introduced in prior work
[Kim et al., NestedNet: Learning Nested Sparse Structures in Deep Neural Networks, CVPR, 2018]
which suggests a nested network to produce multiple different inference paths with different widths (they call "channel scheduling" which is one of their strategies to allow multiple different sparse networks).

Except the missing related work, your paper still has value in terms of different methodology as well as promising experimental results including detection and semantic segmentation.

It would be good to not only introduce additional related work but make the contribution/positioning clear.

---

> ### Author Response · Authors · 2018-12-03
> **Authors' Reply to Comment**
>
> Thanks for your interest in our work! However, we can not fully agree with your suggestion. Our reasons are summarized below:
>
> 1. In your referenced paper [1], the major focus is to compress (Section 3) and sparsify/pruning filters, channels and layers with scheduling (Section 4), and get a "nested sparse networks". The resulted network can be used for model compression, knowledge distillation and hierarchical classification (Section 5).
> In our work, the focus is not to compress, sparsify or pruning, but to simply train a single neural network executable at different width, with the spotlight on the accuracy/performance of standard image recognition benchmarks (ImageNet classification, COCO object detection, instance segmentation, keypoints detection). While the motivation is similar, our focus, methodology, analysis and experimental results are completely different.
>
> 2. Moreover, the only related experiment, hierarchical classification, is also different to our experiments and standard benchmarks. In your referenced paper [1] in Section 5.3:
>
> "We also provide experimental results on the ImageNet (ILSVRC 2012) dataset. From the dataset, we collected a subset, which consists of 100 diverse classes including natural objects, plants, animals, and artifacts."
>
> In efficient deep learning, none of MobileNet v1 [2], MobileNet v2 [3], ShuffleNet [4] evaluate proposed methods on Cifar-10, Cifar-100 or sub-sampled "100-class ImageNet". Many methods that work on toy dataset can not generalize to real scenarios in the topic of efficient models, thus we think challenging settings like standard 1000-class ImageNet is essential to make the work solid and to ensure fair comparisons. Since the motivation is similar, we will be happy to add a citation in related work. We will always be happy to highlight and add comparison to any work that is related and has standard benchmark results.
>
>
> [1] Kim, Eunwoo, Chanho Ahn, and Songhwai Oh. "Learning Nested Sparse Structures in Deep Neural Networks." arXiv preprint arXiv:1712.03781 (2017).
> [2] Howard, Andrew G., et al. "Mobilenets: Efficient convolutional neural networks for mobile vision applications." arXiv preprint arXiv:1704.04861 (2017).
> [3] Sandler, Mark, et al. “MobileNet v2: Inverted residuals and linear bottlenecks: Mobile networks for classification, detection and segmentation." arXiv preprint arXiv:1801.04381 (2018).
> [4] Zhang et al. Shufflenet: An extremely efficientconvolutional neural network for mobile devices.arXiv preprint arXiv:1707.01083, 2017.

---

### Public Comment · ~Bohan_Zhuang1 · 2018-12-10
**Very interesting work, and recommend some related works**

Dear authors,

This is a very interesting work. And I think it is closely related to the mutual learning frameworks [1,2], where the core idea is also to jointly train several models for improving the performance of training each model separately. The main difference is with/without weight sharing, which is one of the contributions of the paper. And I recommend you to cite these works in the paper.

1: Zhang et al. "Deep Mutual Learning", CVPR2018. http://openaccess.thecvf.com/content_cvpr_2018/papers/Zhang_Deep_Mutual_Learning_CVPR_2018_paper.pdf

2: Zhuang et al. "Towards Effective Low-bitwidth Convolutional Neural Networks", CVPR2018
http://openaccess.thecvf.com/content_cvpr_2018/papers/Zhuang_Towards_Effective_Low-Bitwidth_CVPR_2018_paper.pdf

---

> ### Author Response · Authors · 2018-12-10
> **Authors' Reply to Comment**
>
> Thanks for your interest in our work! We will add the citation once revision period is re-opened.

---

### Meta-Review · Area_Chair1 · 2018-12-16
**train a single neural network at different widths**

**Confidence:** 5
**Recommendation:** Accept (Poster)

**Metareview:**

This paper proposed a method that creates neural networks that can run under different resource constraints. The reviewers have consensus on accept. The pro is that the paper is novel and provides a practical approach to adjust model for different computation resource, and achieved performance improvement on object detection. One concern from reviewer2 and another public reviewer is the inconsistent performance impact on classification/detection (performance improvement on detection, but performance degradation on classification). Besides, the numbers reported in Table 1 should be confirmed: MobileNet v1 on Google Pixel 1 should have less than 120ms latency [1], not 296 ms.


[1] Table 4 of https://arxiv.org/pdf/1801.04381.pdf